# Mitochondria and the Repurposing of Diabetes Drugs for Off-Label Health Benefits

**DOI:** 10.3390/ijms26010364

**Published:** 2025-01-03

**Authors:** Joyce Mei Xin Yip, Grace Shu Hui Chiang, Ian Chong Jin Lee, Rachel Lehming-Teo, Kexin Dai, Lokeysh Dongol, Laureen Yi-Ting Wang, Denise Teo, Geok Teng Seah, Norbert Lehming

**Affiliations:** 1Department of Microbiology & Immunology, Yong Loo Lin School of Medicine, National University of Singapore, Singapore 117597, Singaporeteo_shi_hui_rachel@schools.gov.sg (R.L.-T.);; 2Well Programme, Alexandra Hospital, National University Health System, Singapore 159964, Singapore; grace_chiang_sh@nuhs.edu.sg (G.S.H.C.);; 3NUS High School of Mathematics and Science, Singapore 129957, Singapore; 4Department of Cardiology, National University Heart Centre, National University Health System, Singapore 119074, Singapore; 5Division of Cardiology, Department of Medicine, Alexandra Hospital, National University Health System, Singapore 159964, Singapore; 6Chi Longevity, Camden Medical Centre #10-04, 1 Orchard Blvd, Singapore 248649, Singapore; 7Clifford Dispensary, 77 Robinson Rd #06-02, Singapore 068896, Singapore

**Keywords:** unhealthy aging, long COVID, mental neurogenerative disorders, obesity, metformin, gliclazide, GLP-1 receptor agonists, SGLT2 inhibitors

## Abstract

This review describes our current understanding of the role of the mitochondria in the repurposing of the anti-diabetes drugs metformin, gliclazide, GLP-1 receptor agonists, and SGLT2 inhibitors for additional clinical benefits regarding unhealthy aging, long COVID, mental neurogenerative disorders, and obesity. Metformin, the most prominent of these diabetes drugs, has been called the “Drug of Miracles and Wonders,” as clinical trials have found it to be beneficial for human patients suffering from these maladies. To promote viral replication in all infected human cells, SARS-CoV-2 stimulates the infected liver cells to produce glucose and to export it into the blood stream, which can cause diabetes in long COVID patients, and metformin, which reduces the levels of glucose in the blood, was shown to cut the incidence rate of long COVID in half for all patients recovering from SARS-CoV-2. Metformin leads to the phosphorylation of the AMP-activated protein kinase AMPK, which accelerates the import of glucose into cells via the glucose transporter GLUT4 and switches the cells to the starvation mode, counteracting the virus. Diabetes drugs also stimulate the unfolded protein response and thus mitophagy, which is beneficial for healthy aging and mental health. Diabetes drugs were also found to mimic exercise and help to reduce body weight.

## 1. Introduction

When we eat sugar, the levels of glucose in the blood stream rise immediately. In healthy individuals, blood glucose levels are reduced back to normal within two hours by the action of insulin, a peptide hormone produced by the β-cells of the pancreas, which monitor blood glucose levels [1]. Insulin binds to the insulin receptor on the surface of skeletal muscle, fat, and liver cells, stimulating its tyrosine kinase activity [2]. The resulting protein kinase cascade leads to the translocation of the glucose transporter GLUT4 to the plasma membrane and the import of glucose from the blood into the body cells, where glucose is stored as glycogen or fed into glycolysis to generate pyruvate, which can be metabolized to fatty acids or oxidized to carbon dioxide via the krebs (citric acid/TCA) cycle inside the mitochondria [2]. Diabetes mellitus (DM), a human disease characterized by sustained high blood glucose levels, is caused by the β-cells of the pancreas not producing sufficient insulin (type 1; T1DM) or by the cells of the human body developing resistance to insulin (type 2; T2DM) [3]. For the treatment of T1DM, insulin is injected subcutaneously, which allows its gradual absorption into the bloodstream, while T2DM can be treated with oral drugs like gliclazide, which stimulate the pancreas to produce more insulin, and metformin, which restores insulin sensitivity [3]. Metformin has been the main first-line medication for the treatment of T2DM for many decades, but its mode of action is still under debate [4]. At pharmacologically relevant concentrations, metformin inhibits the lysosomal proton pump vATPase, triggering calcium/calmodulin-dependent protein kinase kinase 2 (CaMKK2) to phosphorylate and thereby activate AMP-activated protein kinase (AMPK) at T172 [5]. Metformin also inhibits the mitochondrial electron transport chain, reducing mitochondrial ATP production and increasing cellular AMP levels, which triggers LKB1 to phosphorylate AMPK at T172 as well [6]. Canonically, AMPK is activated, as stated by its name, by high levels of AMP, which result from low levels of ATP during phases of exercise, calorie restriction, and starvation [7]. AMPK switches cells to an energy-conserving program by inhibiting protein biosynthesis and promoting the uptake of glucose and fatty acids [8]. AMPK acts as a heterotrimer, consisting of three different subunits: the catalytic α subunit, the scaffolding β subunit, and the regulatory γ subunit [7]. Humans have two different α subunits, two different β subunits, and three different γ subunits, resulting in 12 different isoforms of AMPK, which phosphorylate and manipulate the activity and proteolytic stability of various protein substrates like transcription factors, ribosomes, and metabolic enzymes to implement energy homeostasis [7]. Different AMPKs are also spatiotemporal and compartmentalized to different subcellular locations like nucleus, cytosol, lysosome, and mitochondria, and they can respond to different signals like the canonical AMP/ATP ratio and post-translational modifications (PTMs) like phosphorylation, ubiquitination, and myristoylation caused by the activation of the respective upstream protein modifying enzymes and the inactivation of the respective protein de-modifying enzymes [9,10,11]. The PTMs work together with the canonical AMP concentration, and they are caused by starvation to essential substrates like glucose and glutamine as well as by cellular stress, including damage to DNA, lysosomes, and mitochondria [7]. Recently, phosphorylation of the AMPK isoforms α1/β2/γ1 and α2/β2/γ1, which are located at the outer mitochondrial membrane and inside the mitochondrial reticulum, was found to be independent of LKB1, which phosphorylates lysosomal AMPK only [11]. These results indicate that more attention could be paid to which AMPK isoforms are activated upon the treatment of human patients with metformin and the other diabetes drugs.

Diabetic medications have off-label health benefits regarding aging, long COVID, mental health, and obesity, and this review aims to elucidate the role of the mitochondria in these health benefits. This review does not cover the off-label health benefits of diabetic drugs for cancer.

## 2. Diabetic Medications: A Key to Longevity Through Mitochondrial Modulation

### 2.1. Introduction into Unhealthy Aging

Unhealthy aging is a progressive, accumulative, and complex phenomenon that involves multifactorial degeneration at cellular, molecular, and organismal levels. Lopez-Otin et al. propose that unhealthy aging is driven by hallmarks that fulfill three criteria [12]: (1) the time-dependent manifestation during the unhealthy aging process, (2) the acceleration of unhealthy aging by experimentally accentuating the hallmark, and (3) the potential to slow down, stop, or reverse unhealthy aging through therapeutic interventions targeting the hallmark [12]. These hallmarks of unhealthy aging are interlinked at the molecular, cellular, and systemic levels and can be classified into three categories: (1) Primary hallmarks (genomic instability, telomere attrition, epigenetic alterations, loss of proteostasis, and disabled macroautophagy) are the fundamental causes of molecular damage associated with unhealthy aging [12,13]. (2) Antagonistic hallmarks (deregulated nutrient-sensing, mitochondrial dysfunction, and cellular senescence) provide beneficial or protective effects at low levels but become harmful at higher levels [12,13,14,15]. (3) Integrative hallmarks (stem cell exhaustion, altered intercellular communication, chronic inflammation, and dysbiosis) emerge when cellular homeostatic mechanisms no longer counterbalance the accumulating damage [12,13].

#### Mitochondria and Unhealthy Aging

Amongst these various factors that lead to unhealthy aging, mitochondrial dysfunction has emerged as a central contributor to unhealthy aging [16]. The mitochondrion is a multifaceted organelle that is integral to regulating cellular metabolism and homeostasis [16]. The mitochondria play a myriad of roles beyond ATP production, such as intracellular signaling, immune response, and cellular redox homeostasis [17]. Mitochondria can create dynamic organellar networks that divide, interact, and fuse with other organelles and guide numerous processes fundamental to cellular health and longevity [18]. The maintenance of cellular health during aging critically depends on the precise balance between mitochondrial quality control processes, particularly the coordination between mitophagy (the selective degradation of damaged mitochondria) and mitochondrial biogenesis [19,20,21].

Mitochondrial dysfunction is associated with multiple aspects of unhealthy aging such as compromised oxidative phosphorylation activity, heightened oxidative damage, deterioration in mitochondrial quality control, diminished activity of metabolic enzymes, and alterations in mitochondrial dynamics, biogenesis, and morphology [22,23]. With unhealthy aging and chronic disease, there is an increase in mitochondria dysfunction via mitochondrial DNA (mtDNA) mutation and depletion [24,25,26,27]; increased reactive oxygen species (ROS); and respiratory chain activities [28]. Altered mitochondria dynamics also affect mitophagy, leading to a buildup of dysfunctional mitochondria [29]. These processes can interplay and hasten the deterioration of tissue function and accelerate the unhealthy aging process. Importantly, mitochondria dysfunction has been identified as a point of convergence for the host of dysregulated pathways in unhealthy aging, T2DM, and obesity [16,27,30]. Research suggests that mitochondrial function actively modulates normal aging, and modulating mitochondrial function or form through interventions such as diet [31] or medications can directly affect organismal longevity [32,33,34,35,36,37]. For instance, recent studies have found that diabetic medications such as metformin, sodium-glucose cotransporter 2 (SGLT2) inhibitors, and glucagon-like peptide-1 (GLP-1) receptor agonists can extend lifespan by targeting the mitochondria (Figure 1) [32,33,34].

This segment examines how diabetic medications may promote longevity beyond glucose control by influencing mitochondrial pathways such as enhancing mitochondrial biogenesis, reducing oxidative stress, and restoring mitochondrial function.

### 2.2. Metformin and Longevity

Metformin was the first diabetic medication to spark interest in using diabetes medications for longevity. Metformin, a biguanide compound, is a first-line medication for the treatment of T2DM. It is effective, has a good safety profile, and is low cost [38]. It improves diabetic control by inhibiting hepatic gluconeogenesis, improving peripheral glucose uptake, enhancing insulin sensitivity, and inducing glycolysis [39]. Early epidemiological studies in diabetic patients showed that individuals taking metformin were living longer and had lower cancer rates than non- diabetic controls, even after accounting for various risk factors, suggesting that its impact extends far beyond its primary role of an antihyperglycemic agent [40,41,42,43]. Subsequent studies have demonstrated that metformin exerts anti-aging effects at both the cellular and organism levels [41]. These effects are mechanistically linked to improvements in key aging markers such as autophagy [44], cellular senescence [45], and inflammation [46]. Preclinical studies have demonstrated extension of lifespan in murine models [32] and model organisms such as *C. elegans* [47]. Metformin has also been shown to protect against many age-related diseases, such as cardiovascular disease, stroke, dementia, and cancer [48,49,50,51,52]. Metformin influences mitochondria function and aging in several interconnected pathways.

#### 2.2.1. Inhibition of Mitochondrial Complex I and AMPK Activation

Unhealthy aging disrupts key nutrient-sensing pathways that respond to alterations in cellular energy status, notably the mechanistic target of rapamycin (mTOR) and insulin/IGF-1 signaling (IIS) pathways [53]. Experimental evidence across species has demonstrated that inhibiting the mTOR or IIS pathway by various means such as caloric restriction or pharmacological intervention extends lifespan [54,55,56,57,58]. Additionally, other nutrient sensors such as AMP-activated protein kinase (AMPK) and sirtuins also become deregulated with age. These pathways contribute to longevity by improving cellular stress resistance and autophagy, which mitigate aging-related diseases [59,60].

Metformin’s primary mechanism of action involves inhibition of mitochondrial complex I of the electron transport chain, leading to alterations in cellular energy status that subsequently activate a cascade of cellular responses that result in the elevation of AMP and the reduction in ATP production, triggering the activation of AMPK [61,62,63,64]. This process parallels responses similar to AMPK activation due to caloric restriction, with downstream effects such as suppressing mTORC1 activity, activating autophagy, and inducing PGC-1α, a co-activator for mitochondrial biogenesis [65,66,67,68,69]. Additionally, metformin inhibits the lysosomal proton pump v-ATPase, which further modulates AMPK activation through LKB1 and CaMKK2-dependent pathways [5]. The cellular reprogramming induced by these combined mechanisms improves metabolic efficiency and stress resistance.

#### 2.2.2. Reduction in Oxidative Stress and Enhanced Antioxidant Capacity

Reactive oxygen species (ROS) are key determinants of aging [70]. ROS accumulation, particularly in mitochondria, is linked to reduced lifespan and premature aging in animals, such as superoxide dismutase-deficient models [71]. This oxidative stress is associated with age-related diseases and the early depletion of hematopoietic stem cells (HSCs) [72], a defect that antioxidants can help reverse. Aging disrupts the balance between ROS production and clearance, increasing oxidants while reducing antioxidant enzymes [73]. Studies on mice have shown that enhancing antioxidant defense, like overexpressing mitochondrial catalase, can combat oxidative damage and extend lifespan [74]. Thus, managing ROS levels is seen as a potential anti-aging approach.

Metformin’s inhibition of complex I alters NADH oxidation and electron flux through the respiratory chain leading to a reduction in ROS production. The resultant redox signaling cascade enhances endogenous antioxidant defense systems, including upregulation of thioredoxin and superoxide dismutase expression in vascular endothelial cells [75,76]. Notably, metformin’s antioxidant effects demonstrate both AMPK-dependent and independent mechanisms, as demonstrated in studies where it restored enzyme activities in animal models under oxidative stress from other drugs [76]. The attenuated ROS production, combined with enhanced antioxidant capacity, creates a more favorable redox environment that may contribute to cellular longevity.

#### 2.2.3. Mitochondrial Quality Control and Longevity

Mitochondrial quality control functions through the coordination of various mechanisms (proteostasis, biogenesis, dynamics, and mitophagy) and is core to maintaining normal mitochondria morphology and performance [77,78]. It prevents accumulation of ROS [79] and clears damaged mitochondria through mitochondrial-derived vesicles (MDVs) [80], mitochondrial proteases, mitochondrial unfolded protein response (UPRmt) [81], and mitochondrial mitophagy/autophagy [80]. Mitochondrial quality control regulates mitochondrial homeostasis and prevents abnormal polypeptides from entering the mitochondria during mitochondria biogenesis [78]. Mitochondrial dysfunction when exacerbated by failing mitochondrial quality control processes accelerates abnormal energy metabolism and mitochondrial dysfunctional-induced senescence—a major contributor to unhealthy aging [78,82].

Metformin beneficially impacts on mitochondrial quality control by enhancing mitophagy and stimulating mitochondrial biogenesis [83,84,85,86,87,88]. It triggers a hormetic response to mitochondria resulting in a series of adaptive cellular processes, involving the upregulation of key transcriptional regulators, notably PGC-1α and nuclear factor erythroid 2-related factor 2 (Nrf2), which orchestrate AMPK-mediated mitochondrial biogenesis and amplify the integrative anti-inflammatory response network that enhances longevity [83,86,87,88,89,90]. Mitophagy, a critical mitochondria quality control mechanism, aids in clearing damaged mitochondria and oxidized by-products [91]. This process of mitochondrial recycling is crucial for maintaining healthy mitochondria, essential for cellular energy production [91]. One such pathway involved in mitophagy relies on the proteins, PINK1 and parkin, which are induced by metformin [83]. Dysfunctional mitochondria accumulate PINK1 on their surface, recruiting parkin to initiate the degradation process [92]. Additionally, metformin stimulates mitochondrial biogenesis [83,86,88]. This is achieved by activating key transcription factors like PGC-1α, which orchestrates the expression of genes involved in mitochondrial growth and function [68,69].

UPRmt helps to counter mitochondrial stress by inducing the activation of antioxidant enzymes and mitochondrial chaperones/proteases [93], which remove damaged polypeptides and regulate protein processing, maturation cleavage, and assembly [94]. A recent study in rats indicated that metformin improves mitochondrial dynamics and biogenesis by regulating UPRmt and increasing the mtDNA/nDNA ratio [95]. Metformin may offset the decline in mitochondrial function by promoting both mitophagy and biogenesis; these processes selectively eliminate damaged mitochondria while promoting the proliferation of healthy organelles, ultimately resulting in a more robust mitochondrial network [78,91].

#### 2.2.4. Metabolic Reprogramming and Longevity

At high concentrations, metformin inhibits cellular oxygen consumption rate and triggers metabolic reprograming by enhancing glutaminolysis and glycolysis [90]. Following metformin’s inhibition of mitochondrial respiratory complex I, the altered mitochondrial redox state leads to significant changes in pyruvate metabolism and related pathways [6,96]. This results in a shift in substrate utilization away from pyruvate oxidation and increases the cell’s reliance on alternative substrate oxidation pathways, namely fatty acid oxidation, resulting in metabolic flexibility and improved mitochondrial respiratory efficiency [89]. These adaptations ultimately improve oxidative mitochondrial capacity and enhance biogenesis and mitophagy while reducing mitochondrial dysfunction [90,97].

#### 2.2.5. Genomic Stability and Longevity

DNA damage caused by both external factors (i.e., ultraviolet light, ionizing radiation, and genotoxic chemicals) and internal factors (i.e., products of cellular metabolism) is a major factor in unhealthy aging [98]. This accumulation of DNA damage progressively impairs transcription and replication, thus increasing cellular senescence and apoptosis, which are key hallmarks of unhealthy aging [99].

By inhibiting damage to mitochondrial and nuclear DNA and scavenging free radicals, metformin has demonstrated potential in facilitating DNA repair and preventing DNA damage. For instance, metformin has been shown to prevent insulin-induced DNA damage in cells [100] and to inhibit radiation-induced DNA damage in bone marrow hematopoietic stem cells in murine models [101]. Metformin also supports p53-mediated DNA repair by activating AMPK through the inhibition of mitochondrial complex I-mediated oxidative phosphorylation and reduction in cellular ATP levels [98,102]. Notably, metformin’s ability to mitigate mitochondrial genomic instability via reduction in oxidative stress, scavenging mitochondrial ROS, only requires one tenth the concentration of metformin required to inhibit mitochondrial complex I [72].

### 2.3. GLP-1 Receptor Agonists and Aging

GLP-1 receptor agonists have not only proven to be effective in managing T2DM, obesity [103], cardiovascular risk [104], and renal dysfunction [105] but are now also emerging as a potential drug for improving health span and reducing age-related diseases [34,106,107]. Findings from preclinical studies have suggested that GLP-1 receptor agonists may counter cellular changes that are associated with unhealthy aging by ameliorating mitochondrial dysfunction [108], reducing oxidative stress [109], limiting cellular senescence [110], and protecting against chronic inflammation [111].

#### 2.3.1. Mitochondrial Biogenesis and Function

GLP-1 receptor agonists have been associated with increased mitochondrial biogenesis and improved mitochondrial function, which are both integral for ATP production [112]. Liraglutide and exenatide support mitochondrial health through various processes, with liraglutide ameliorating mitochondrial dysfunction via the cAMP/PKA pathway [108] and protecting cardiomyocytes from interleukin-1β-induced mitochondrial dysfunction [113]. GLP-1 receptor activation in H9c2 cardiomyoblasts also helps to reduce methylglyoxal-induced mitochondrial dysfunction [114]. Similarly, exenatide improves mitochondrial structure and dynamics and rectifies mitochondrial energy crisis [115].

#### 2.3.2. Reduction in Oxidative Stress

GLP-1 receptor agonists may be able to lengthen health span through a multifaceted process that inhibits cell senescence and maintains genomic integrity by reducing oxidative stress [109,116,117,118,119]. This protection is mediated in part due to GLP-1 receptor activation-related reduction of ROS production and inflammation and protecting against apoptosis by engaging SIRT1 [120].

### 2.4. Sodium-Glucose Cotransporter 2 (SGLT2) Inhibitors and Longevity

SGLT2 inhibitors modulate nutrient sensing pathways such as AMPK, mTOR, and the inflammasome, mimicking key effects of caloric restriction [33,121,122,123]. SGLT2 inhibitors also increase natriuresis and sodium delivery to the distal nephron, normalizing the tubuloglomerular feedback mechanism and improving blood pressure, which is integral for cardio-renal protection [124,125], thus slowing or preventing common long-term renal, cardiovascular, and metabolic complications such as chronic kidney disease, heart failure, atrial fibrillation, atherosclerotic disease, and all-cause hospitalization [126,127,128,129,130]. There is a large body of evidence that indicates that SGLT2 inhibitors reduce the risk of many age-associated diseases and improve life expectancy despite the absence of diabetes as a central pathology [33,131,132]. Given the benefits of organ protection and long-term health outcomes, SGLT2 inhibitors have now emerged as class I recommendations in the guidelines for treatment of cardiac and renal disease irrespective of diabetes status [132,133].

SGLT2 inhibitors reduce glucose by inhibiting glucose reabsorption in the proximal tubules promoting glucosuria, leading to caloric loss and a subsequent metabolic shift at the systemic level towards ketone and fatty acid utilization, thought to be cardio and renal protective [124,125]. This shift to reduce cellular lipotoxicity and improve tubuloglomerular feedback stimulation also reduces cellular glucotoxicity and oxidative stress, improving mitochondrial activity [124,125]. SGLT2 inhibitors as caloric restriction mimetics trigger off a cascade of interlinked processes such as promotion of autophagy, cellular integrity optimization, inhibition of apoptosis, reduction in oxidative and endoplasmic reticulum stress, mitochondrial biogenesis, restoration of mitochondrial health, and reduction of inflammatory and profibrotic pathways, which in turn induce senolysis and inhibit the accumulation of senescent cells [33,123]. The beneficial mechanisms of action of SGLT2 inhibitors are now increasingly recognized as possible novel therapeutic agents for longevity [33,121,122,123]. With regard to the mitochondrial function-mediated benefits of SGLT2 inhibitors, they improve mitochondrial energetics and oxidative defense through their natriuretic effects and modulate mitochondrial dynamics through the regulation of mitochondrial fusion and fission [33,131,134].

#### 2.4.1. Mitophagy and Cellular Rejuvenation

SGLT2 inhibitors promote autophagy via mechanisms similar to caloric restriction. Activation of autophagy leads to cellular digestion of damaged oxidized proteins and dysfunctional organelles for cellular restoration. Specifically, SGLT2 inhibitors stimulate mitophagy, promoting the recycling of damaged mitochondria and supporting mitochondrial biogenesis [135], ultimately restoring ATP production in energy-depleted cells [134,136]. SLGT2 inhibitors also support cellular longevity by decreasing ROS levels and improving mitochondrial integrity through reducing markers of oxidative stress such as dysfunctional mitochondrial and endoplasmic reticulum stress [137,138]. By augmenting an autophagic influx, SGLT2 inhibitors also enhance energy production and mitochondrial biogenesis through the activation of longevity pathways such as AMPK, SIRT1, SIRT3, SIRT6, and PGC1-α [131].

#### 2.4.2. Modulation of Mitochondrial Dynamics and Longevity

Growing evidence suggests that SGLT2 inhibitors may modulate mitochondrial dynamics [134]. Mitochondrial dynamics are linked to various cellular processes, such as cell cycle, apoptosis, cell migration, mitophagy, and ROS production [139]. Mitochondrial fission and fusion, the integral processes of mitochondrial dynamics, determine mitochondrial morphology and are critical for functional maintenance of mitochondria. Mitochondrial fusion can help compensate for mitochondrial deficits and extend mitochondrial lifespan, while mitochondrial fission enhances mitochondrial quality control by segregating damaged mitochondria and facilitating mitophagy [140,141]. A fine balance between the two is crucial for both cell survival and functioning.

Ipragliflozin reduces mitochondrial dysfunction caused by high-fat diets by restoring key regulators of mitochondrial fusion, optic atrophy 1 (Opa1) and mitofusin 2 (Mfn2), to normal values [142]. Likewise, dapagliflozin augments mitochondrial function by normalizing mitofusin 1 (Mfn1) and Mfn2, preserving mitochondrial membrane potential [143]. Excessive mitochondrial fission causes membrane permeabilization, cristae disorganization, and release of pro-apoptotic proteins, contributing to cellular death [144]. Empagliflozin restores the AMP-to-ATP ratio leading to AMPK activation, triggering dynamin-related protein 1 (Drp1), a critical effector of mitochondrial fission [145]. Drp1S637 phosphorylation is increased while Drp1S616 phosphorylation is decreased, suppressing mitochondrial fission [145]. Empagliflozin-induced inhibition of mitochondrial fission suppresses mitochondrial ROS oxidative stress to delay cardiac microvascular endothelial cell (CMEC) senescence [145]. Via this cascade of AMPK-mediated inhibition of mitochondrial fission, empagliflozin serves to improve diabetic cardiac microvascular injury and prevent heart failure [134,145]. Through these mechanisms, SGLT2 inhibitors aid in revitalizing organ function, preventing adverse structural remodeling [135,146], and sustaining normal tissue architecture [147], which might lead to an overall improvement in life expectancy and health span.

### 2.5. Discussion for Longevity

The emerging evidence connecting diabetic medications to longevity represents a paradigm shift in our understanding of these therapeutic agents. While their glucose-lowering effects have been well established, their impact on mitochondrial function and subsequent influence on unhealthy aging processes present new horizons in therapeutic applications. Metformin’s ability to activate AMPK and mirror benefits of caloric restriction, modulating mitochondrial efficiency and reducing oxidative stress while maintaining adequate ATP production, suggests a delicate balance between energy metabolism and cellular longevity [32,35]. The concurrence of molecular mechanisms and long-term clinical observations has positioned metformin as a leading candidate for targeting the unhealthy aging process, though important questions remain about optimal dosing, timing, and population-specific effects in non-diabetic individuals.

Other classes of diabetic medications such as SGLT2 inhibitors and GLP-1 receptor agonists have shown promising results in improving lifespan [33,34,148]. SGLT2 inhibitors’ anti-aging mechanisms of actions such as their ability to modulate mitochondrial dynamics, upregulate nutrient deprivation signaling, and downregulate nutrient surplus signaling slow cellular aging by augmenting autophagy [33]. GLP-1 receptor agonists may also influence aging mechanisms by their ability to activate multiple intracellular signaling pathways such as the cAMP/PKA pathway and other pathways associated with anti-inflammatory and antioxidative stress responses [148].

Overall, the evidence supporting the role of diabetic medications in improving cellular resilience by improving mitochondrial health and enhancing mitochondrial biogenesis is compelling [68,69,108,113,114,131,135,137,138,148].

Given the potential lifespan-enhancing effects of these diabetic medications, there is a role for these medications to be repurposed as longevity therapeutics in non-diabetic individuals. However, current research is limited by the paucity of long-term randomized controlled trials especially tailored to assess for longevity outcomes, variability in mitochondrial function assessment methods, and a dearth of information regarding the complex interconnection between various metabolic pathways affected by diabetic medications. Moreover, further research is required in human studies to review how the role of patient-specific factors such as age, gender, comorbidities, and epigenetics determine treatment or dose response.

Presently, there are two major clinical trials, the Metformin in Longevity Study (MILES) [149] and the Targeting Aging with Metformin (TAME) [41] study, investigating the potential anti-aging effects of metformin in non-diabetic populations. Analysis of data from MILES reveals that metformin may cause anti-aging transcriptional changes; however, its findings are unable to identify the primary site of action of metformin and will need to be validated in other tissues and study designs [149]. TAME may provide further insight on whether metformin reduces the risk of developing age-associated diseases in non-diabetic individuals; however, there are both reservations regarding the age-dependent effects of metformin [41] and concerns about the variable effects of metformin in humans and organisms (*C. elegans*, rodents) [150]. Nonetheless, findings from these studies will be useful to further our assessment of the benefits of metformin on health span and in improving future trials. Better understanding of the mechanisms through which diabetic medications attenuate the hallmarks of aging such as mitochondrial dysfunction can facilitate the development of novel and effective strategies that directly target the biology of aging.

## 3. Involvement of Mitochondria in the Pathogenesis and Treatment of Metabolic Disorders of COVID-19 and Long COVID

### 3.1. Introduction into Long COVID

Post-acute sequelae of SARS-CoV-2 (PASC), also known as “long COVID”, is currently defined as an infection-associated chronic condition that occurs after SARS-CoV-2 infection and is present for at least 3 months as a continuous, relapsing and remitting, or progressive disease state that affects one or more organ systems [151]. It can follow asymptomatic, mild, or severe SARS-CoV-2 infection and can exacerbate preexisting health conditions or present as new conditions. A conservative estimate suggests that long COVID affects 10% of COVID-19 patients [152]. Long COVID disease manifests in multiple ways, affecting any organ system. There may be single or multiple symptoms (e.g., persistent fatigue, breathlessness, headaches, memory disturbances, rapid heart rate, muscle weakness). Long COVID could manifest as single or multiple diagnosable clinical conditions (e.g., diabetes, clotting disorders, anxiety, migraine, stroke, cardiovascular disease, autoimmune diseases) [151]. Long COVID can affect anyone, but factors predicting greater risk include female sex, non-vaccinated patients, repeated infection, and more severe infection [153,154]. A year after acute COVID-19 infection, circulating spike antigen is detectable in 60% of those with symptoms of long COVID but in none of the healthy COVID-19 survivors, suggesting a link between viral persistence and long COVID disease manifestations [155]. The specific human reservoirs of the virus remain unclear, but persistent gut shedding of SARS-CoV-2 proteins has been documented [156].

The leading theories for the pathogenesis of long COVID invoke immune, metabolic, and/or endothelial dysfunction. This part of the review will focus on the aspects of mitochondrial derangements, particularly those affecting glucose metabolism, which are associated with COVID-19 infection and could lead to long COVID. Potential mitochondria-targeted therapies based on natural products or repurposing existing drugs will be discussed.

### 3.2. Metabolic Derangements in Long COVID

Multiple long-term metabolic derangements (hyperlipidemia, poorer glycemic control, weight gain, and persistently elevated liver enzymes) persist 6 months after non-severe COVID-19 infection [157]. A meta-analysis shows that COVID-19 is associated with a 66% higher risk of new-onset diabetes post-infection (risk ratio, 1.66; 95% CI 1.38; 2.00) [158]. The excess burden of post-COVID diabetes is 13.46 per 1000 people at 12 months after acute infection [159]. This is unlikely to be simply due to viral infection of pancreatic islet cells [160]. A potential reason for the higher diabetic risk is insulin resistance after infection [161], due to adipose tissue dysfunction or downregulation of ACE2. Increase in Golgi protein 73 (GP73), free fatty acids (FFA), phosphoenolpyruvate carboxykinase (PEPCK), and hyperinsulinemia triggered by insulin resistance drive hepatic gluconeogenesis, leading to hyperglycemia [162]. GP73 is a glucogenic hormone contributing to SARS-CoV-2-induced hyperglycemia [163], while free fatty acids have been linked to mitochondrial dysfunction [164]. There are two 63% identical isoforms of PEPCK, which catalyze the rate-limiting step in gluconeogenesis in the liver: a cytosolic form (PCK1) and a mitochondrial form (PCK2). In primary hepatocytes infected with SARS-CoV-2, a three-fold increase in PEPCK enzymatic activity was measured without a corresponding increase in PCK1 mRNA levels [165]. The mRNA level of PCK2 was not determined, and Western blot was not performed [165], which means that it remains unclear if SARS-CoV-2 infection increases the mRNA levels of PCK2 or the specific enzymatic activity of PCK1 by changing its post-translational modifications like S90 phosphorylation [166] or K91 acetylation [167]. Gluconeogenesis takes place inside the cytosol, which might explain why the authors failed to look at PCK2, which resides inside the mitochondria; however, there is a PEP carrier in the mitochondrial membrane, and both PEPCK isoforms were shown to be involved in the generation of glucose from oxaloacetate via PEP [168]. It has also been suggested that new onset diabetes post-COVID is because increased glycolysis during COVID-19 infection [169,170] increases methylglyoxal, which contributes to diabetes via multiple mechanisms, including pancreatic endothelial cell dysfunction and impairment of insulin signaling pathways [171].

### 3.3. SARS-CoV-2 Affects Host Mitochondria

Since the start of the COVID-19 pandemic, it has been recognized that SARS-CoV-2 hijacks the host’s mitochondria, and this accounts for much of the viral-induced pathology [172,173,174,175]. Besides structural damage to mitochondria [176], the expression of genes associated with mitochondria function is altered in long COVID patients [177], and such patients have elevated markers of mitochondria dysfunction and oxidative stress [178]. Increasing disease severity is associated with decreasing mitochondria DNA copy numbers, which suggests mitochondria fusion [179].

High levels of glycolysis induced by SARS-CoV-2 in immune cells and lung epithelial cells lead to a “Warburg-like” metabolic shift to favor aerobic glycolysis, with increased glucose uptake, which supports viral replication [169] but reduces ATP production [170,171]. Hence, despite SARS-CoV-2 ramping up glycolysis in the host cell [169,170], the mitochondria appear to have significantly reduced ATP-linked respiration, reserve capacity, and maximal respiration—all consistent with a compromised mitochondrial function—in peripheral blood mononuclear cells of COVID-19 patients, but not patients with chest infection from other causes [180]. This is corroborated by mitochondrial stress test analysis showing a marked disruption in oxidative phosphorylation, induced by expression of viral proteins N, ORF3a, and NSP7 in bronchial epithelial cells [181]. High arterial lactate at low exercise levels noted in long COVID patients suggests premature transition from fatty-acid oxidation to carbohydrate oxidation, which implies mitochondrion metabolic reprogramming by the virus [182]. In studies of nasopharyngeal samples from COVID-19 patients, expression of mitochondrial genes involved in energy production is suppressed in the nasopharynx during acute infection. Specifically, there is lowered transcription of certain nuclear DNA-encoded mitochondrial genes affecting inner membrane transport enzymes responsible for correct functioning of the TCA cycle and oxidative phosphorylation. Notably, repression of these mitochondrial genes in the heart, kidney, liver, and lymph nodes is demonstrated in autopsied patients long after the virus has apparently been cleared from the body [183]. Hence, SARS-CoV-2 can cause long-lasting damage to energy production by mitochondria in multiple organs, and this explains the chronic fatigue and muscle weakness often experienced by long COVID patients.

SARS-CoV-2 dsRNA (produced during viral replication) has been found within mitochondria, where it impairs mitophagy [184,185]. Failure to degrade damaged mitochondria leads to excessive reactive oxygen species (ROS) formation [185], causing oxidative stress in patients, which may be another basis for the pathology in long COVID. Mitochondria are a major source of ROS such as H_2_O_2_ and superoxide in the cell, which arise from disruptions to the electron transport chain, making them extremely vulnerable to oxidative stress. Mitochondrial dysfunction and increased levels of mitochondria-derived ROS can activate NF-κB and the NLRP3 inflammasome [186]. The hyperglycolytic state also promotes production of pro-inflammatory cytokines and ROS [170]. The ensuing chronic inflammation perpetuates tissue damage in multiple organs, accounting for many symptoms of long COVID [187]. The above illustrates why mitochondrial dysfunction is a hallmark of COVID-19 and long COVID pathogenesis.

### 3.4. Hyperglycemia Associated with Mitochondria Dysfunction

Hyperglycemia has long been known to be linked to mitochondrial dysfunction, through increased ROS generation and associated mitochondrial fragmentation [188]. Reduced expression of peroxisome proliferator-activated receptor gamma coactivator-1α (PGC-1α), the master transcriptional regulator of mitochondrial biogenesis, is noted in insulin-resistant patients [189]. Type 2 diabetics have impaired skeletal muscle mitochondria oxidative phosphorylation [190]. These pieces of evidence show that persistent hyperglycemia is associated with impaired mitochondrial oxidative capacity and/or mitochondrial function, and conversely, mitochondria dysfunction has a key role in insulin resistance [191]. Collectively, they can explain why mitochondrial dysfunction caused by SARS-CoV-2 [192] may be a major cause of persistent hyperglycemia in long COVID patients.

### 3.5. Mitochondria Targeting in COVID-19 Treatment

Several studies have demonstrated that obesity increases the risk of long COVID [193,194]. A phase 3 clinical trial [195] showed that two-week treatment of obese patients during acute COVID-19 infection with a common diabetes drug metformin reduced their risk of long COVID by day 300 by 41.3% compared with a placebo, with an estimated cumulative incidence of 6.3% in the metformin group and 10.6% in the placebo group. We speculate that a potential reason for the effectiveness of metformin in this context is as follows. The enzyme 5′adenosine monophosphate-activated protein kinase (AMPK) is a metabolic sensor that detects cellular energy levels and acts to stimulate processes that increase ATP, such as glycolysis and fatty acid oxidation, while inhibiting ATP-utilizing processes such as gluconeogenesis [196]. Metformin is a biguanide that induces PGC-1α expression and also indirectly activates AMPK by inhibiting complex I of the mitochondrial respiratory chain, leading to an increased AMP:ATP ratio [5]. AMPK promotes mitochondrial biogenesis via activation of PGC-1α but also promotes mitochondrial fission to ensure appropriate mitochondrial turnover for homeostasis [197]. Metformin induction of AMPK-dependent mitochondrial fission improves mitochondrial respiration and restores the mitochondrial life cycle in vivo, correlating with the AMPK-dependent blood glucose-lowering effect of metformin in mice fed with a high-fat diet [198]. By extension, perhaps other drugs that directly or indirectly target mitochondria metabolic dysfunction may be useful in preventing or treating metabolic symptoms of long COVID.

Some of these mitochondria-targeting drugs are natural products that activate PGC1-α, AMPK, or nuclear factor erythroid 2–related factor 2 (Nrf2). However, the caveat is that most natural products have multiple mechanisms of action, so it may be difficult to distinguish their effects on the mitochondria from other beneficial effects. Sulforaphane, resveratrol, and curcumin activate Nrf2. Nrf2 regulates mitochondria ROS formation, promotes mitophagy, promotes mitochondrial biogenesis and more efficient oxidative phosphorylation, and also protects mitochondria from oxidative stress by increasing transcription of antioxidants [199]. Separately, Nrf2 also reduces inflammation through repression of transcriptional upregulation of pro-inflammatory cytokine genes [200], and it is mainly for this purpose that Nrf2 activators are currently used clinically. However, in a recently published clinical trial, the most widely used Nrf2 activator dimethyl fumarate (DMF) did not alter the clinical status of hospitalized COVID-19 patients [201]. Arguably, in this study, the 5-day observation period for each patient does not allow adequate time for long-term Nrf2 effects on mitochondria to be clinically evident; hence, performing a DMF trial as treatment for long COVID instead, over at least 6 months, would still be relevant. Likewise, a short-term resveratrol trial for treating mild COVID-19 cases in an outpatient setting also showed no significant benefit [202], but the logistics of that study led to a median lag time of 5 days from symptom onset to treatment, which would most likely have reduced the therapeutic benefit, and its effect on long COVID was not studied. Quercetin (a flavonoid) and hydoxytyrosol (a polyphenol) improve mitochondrial biogenesis through activation of PGC-1α via AMPK [199]. Clinical studies on these products have been very small in scale [203] and focused mainly on speed of recovery from acute COVID-19 symptoms, but their effects on mitochondria pathology deserve longer-term in-patient study.

Very few studies of COVID-19 treatment have focused on reversing viral-induced metabolic changes in mitochondria. 2-deoxy-D-glucose (2DG) represses glycolysis and stimulates respiration, directly opposing the effect of SARS-CoV-2 on human cells. 2DG was given emergency approval for clinical use to treat COVID-19 in India in 2021 after a successful clinical trial but is no longer in active usage after the advent of anti-viral treatment. Murine studies showed that the main effect of an extended, intermittent 2DG treatment on mice is to augment the mitochondrial respiratory chain proteome in the heart, implying an increase in mitochondrial oxidative capacity [204]. Studies on extended, intermittent 2DG clinical usage would thus be relevant in the context of long COVID treatment. SARS-CoV-2 activates the epidermal growth factor receptor (EGFR) signal cascade and enhances the mitochondrial localization of EGFR during the early stage of infection, leading to increased ATP production [205]. Existing FDA-approved EGFR inhibitors such as vandetanib reverse the above processes, restore SARS-CoV-2-induced abnormal upregulation of most genes that modulate oxidative phosphorylation, and reduce viral propagation. Hence, drugs that alter mitochondrial bioenergetics have potential for use in SARS-CoV-2 treatment [205], so clinical trials on vandetanib against long COVID would be justified (Figure 2).

### 3.6. Discussion for Long COVID

It has become clear that strategies that work well in treating acute COVID-19, such as anti-viral therapy, may fail to benefit long COVID patients, as evidenced by the disappointing outcome of the STOP-PASC trial involving the nirmatrelvir-ritonavir combination [206]. It is likely that the converse is also true—drugs that are effective against long COVID may fail to speed up recovery from acute infection, so efficacy in resolving acute COVID-19 cases should not be the absolute criterion to filter potential candidate therapies for long COVID. Severe pathology in acute infection is often due to the cytokine storm causing overwhelming inflammation, but the pathogenesis of long COVID involves long-term defects in energy production and metabolism of sugars, proteins, and fatty acids, with additional damage from ROS. Despite a wealth of knowledge on mitochondria derangements associated with COVID-19 and long COVID, it is curious that none of the clinical trials currently underway directly aim to address mitochondria metabolic defects. The majority of the trials that involve pharmacological agents aim to reduce inflammation or ameliorate neurological, cardiac, or respiratory symptoms [207]. Given that long COVID manifests as a myriad of diverse clinical conditions and symptom complexes [151], organ-specific therapies would benefit only a subset of long COVID patients. More focus on interventions aimed at treating the underlying pathophysiology of long COVID is warranted, as managing cellular metabolic derangements has the potential to yield benefits in multiple organs.

## 4. Diabetic Medications: A Key to Mental Health Through Mitochondrial Modulation

### 4.1. Alzheimer’s Disease

#### 4.1.1. Introduction to Alzheimer’s Disease

Alzheimer’s disease (AD) is a progressive neurodegenerative disease [208]. Initially, AD patients exhibit symptoms of mild cognitive impairment (MCI), where they experience problems with memory and thinking [208]. As the disease advances, MCI progresses into dementia and could also be accompanied with motor impairments in severe AD [208]. AD is typically characterized by the accumulation of amyloid-β (Aβ) plaques and tau neurofibrillary tangles in the brain, and these two protein aggregates have been thought to contribute to the pathophysiology of AD [209,210,211,212].

#### 4.1.2. Mitochondria and Alzheimer’s Disease

Besides Aβ and tau, mitochondria have also been shown to be implicated in AD. Abnormal mitochondrial morphology has been observed in the neurons and fibroblasts of AD patients [213,214,215], and accumulation of Aβ was also observed in the mitochondria of transgenic AD mice models [216,217]. Mitochondrial dynamics involving mitochondrial fission, fusion, and mitophagy have also been found to be impaired in AD. Studies have shown that alterations in mitochondrial fission and fusion protein levels lead to abnormal mitochondrial dynamics, although contrasting results have been observed regarding the exact alteration (increase/decrease) for specific proteins like dynamin-related protein 1 (DRP1) [214,218,219,220]. Similarly, impaired mitophagy has also been observed [221,222,223].

Beyond impaired mitochondrial dynamics, increased oxidative stress and decreased mitochondrial metabolism due to mitochondrial dysfunction are also involved in AD. Measurements of lipid, nucleic acid, and protein oxidative markers indicated increased oxidative stress in both MCI and AD patients [224,225,226,227]. Oxidative stress has been proposed to be present at the early stages of AD development prior to Aβ plaque accumulation [228,229], but Aβ plaques can induce oxidative stress as well [230,231,232]. This increase in oxidative stress can eventually cause apoptosis and neuronal cell death, thereby contributing to AD pathogenesis [233,234,235,236,237].

Altered metabolism is also linked to AD. Hypometabolism of glucose has been observed in MCI and AD patients when compared to normal controls [234,238,239]. Similarly, a decrease in cerebral glucose uptake or metabolism was also observed in AD mice models [240,241]. Furthermore, it was also found that the restoration of glucose metabolism was able to improve cognition in AD mouse models [242]. In addition to that, the mitochondrial metabolic pathways in AD could also be altered in AD. Transcriptomics and proteomics studies have revealed a downregulation of several genes that are part of the TCA cycle in AD [243,244]. This downregulation was also observed for nuclear encoded genes involved in oxidative phosphorylation, but some studies did not observe the same results for mitochondrial-encoded oxidative phosphorylation genes [245,246,247,248,249]. Beyond metabolic processes, the downregulation of glucose transporters GLUT1 and GLUT3 were also observed in AD patients [250,251,252].

To further support the importance of altered metabolism in AD, insulin resistance has also been found to be related to AD as well. AD has been proposed to be termed as “Type 3 Diabetes” [253]. Meta-analysis and cohort studies have shown that patients with Diabetes mellitus (DM) have a higher risk of dementia, which includes AD [254,255,256,257,258]. Diabetic mouse models have also been shown to exhibit AD-like changes [259].

Anti-diabetic drugs have been tested as potential therapeutics for AD. One such anti-diabetic drug that has been investigated is metformin. Metformin has been investigated using various experimental methods to determine its efficacy on AD. Metformin has been found to improve AD-related pathological changes in a neuronal cell line with prolonged exposure to insulin [260]. In addition to that, metformin has also been found to improve cognitive functions of AD mice models [261,262,263,264]. Several studies in mice have also shown that metformin reduces the amount of Aβ and phosphorylated tau [262,263,264]. The effect of metformin on general dementia and AD has also been investigated in observational studies and clinical trials. Cohort studies have been done to investigate the risk of dementia in Type 2 DM patients with and without metformin treatment [265,266,267,268]. These non-metformin patients were either given other anti-diabetic medication or treated without any pharmacological interventions. The results identified that patients treated with metformin had a lower risk of developing dementia compared to those treated without [265,266,267,268]. Furthermore, a nested case control study has also found that long-term metformin usage was also associated with a lower risk of AD [269]. A double-blinded randomized control cross-over study conducted for patients with AD-related mild MCI or mild dementia found that executive function was significant improved with metformin use [270].

Metformin could be exerting its effect through improving mitochondrial-related abnormalities observed in AD. Metformin has been found to improve mitochondrial morphology and reduce oxidative stress in AD models [263,271,272]. One way this could be done is through the AMP activated protein kinase (AMPK)-dependent effects of metformin. Metformin has been known to activate AMPK [273], and metformin administration has been found to increase phospho-AMPK levels in mice modeling AD [263,264]. In a rat AD model, metformin has been shown to improve memory in an AMPK-dependent manner, as metformin treatment reduces the escape latency times of AD-induced rats, but the addition of the AMPK inhibitor dorsomorphin increases the escape latency time of the metformin-treated rat [274].

However, there are also many studies that have suggested that metformin has a negative impact on AD. Some studies have suggested that metformin increases Aβ generation and impairs cognition in AD mice models [275,276]. Beyond this, some cohort studies have also shown that the use of metformin is associated with an increased risk of dementia or AD in diabetic patients [277,278,279]. These contrary results suggest that more research has to be done to ascertain the benefits of metformin in AD. With multiple clinical trials currently in progress (NCT04511416, NCT05109169, NCT04098666), more might be revealed about the possibility of using metformin in AD treatment.

Besides metformin, other anti-diabetic drugs have also been tested. Limited evidence has been found regarding the use of gliclazide in AD. Gliclazide has been found to exhibit neuroprotective effects in diabetic rats [280,281], and one study has suggested that gliclazide has been shown to have antioxidant effects [280]. Besides that, research on *C. elegans* has shown that a palladium coordination compound containing gliclazide has been found to have protective effects on Aβ-induced toxicity, suggesting a possible benefit in AD [282]. Besides that, sulfonylureas, which is the class of drug that gliclazide is part of, has been investigated for its association with dementia in multiple cohort studies. While some have suggested that sulfonylureas are associated with a lower risk of dementia [283], others have concluded otherwise [284,285]. These limited and contradictory data suggest that more research is required to investigate the role of gliclazide in AD.

The SGLT2 inhibitors canagliflozin and empagliflozin have also been investigated in AD. Canagliflozin has been found to improve cognitive deficits and memory in streptozotocin and scopolamine-induced AD mice models [286,287], accompanied by reduction in tau and Aβ [286]. While it has been suggested that canagliflozin exerts its effect through the inhibition of acetylcholinesterase [287], it has also been shown that canagliflozin can reduce oxidative stress through the AMPK pathway [286]. Similarly, a decrease in Aβ plaques and improved cognitive function were also observed in an AD mice model upon empagliflozin treatment [288].

Several cohort studies have also shown the potential of SGLT2 inhibitors as a treatment for AD. The use of SGLT2 inhibitors has been found to be associated with higher cognitive scores in diabetic patients [289]. In addition to that, a nested case control study has shown that SGLT2 inhibitors were associated with lower odds of dementia [290]. Population-based studies have also shown that patients using SGLT2 inhibitors have lower risk of dementia when compared to non-SLGT2 inhibitor users [192,291,292]. Beyond that, patients using SGLT2 inhibitors were found to have a lower risk of dementia when compared to those taking DPP4 inhibitors [293,294]. While most of these studies investigated SGLT2 inhibitors as a whole, one study that looked at the association of individual SGLT2 inhibitors found that empagliflozin showed moderate reduction in risk for dementia, while canagliflozin showed no association [293]. In terms of clinical trials, one clinical trial has found that the use of incretins and SGLT2 inhibitors did not cause a reduction in cognitive function in diabetic patients [295], while another clinical trial has recently been completed (NCT05081219).

### 4.2. Parkinson’s Disease

#### 4.2.1. Introduction to Parkinson’s Disease

Parkinson’s disease (PD), described initially by James Parkinson [296], is a progressive neurodegenerative disorder exhibiting various movement disabilities such as bradykinesia, tremor, and rigidity, although non-motor symptoms can be present as well [297,298]. PD is typically characterized by the accumulation of Lewy bodies and neurites consisting of misfolded α-synuclein (α-syn) [298]. Although PD has always been thought to be caused by a loss in dopaminergic neurons in the substantia nigra, it has also been shown that PD also affects other areas of the peripheral and central nervous system [298].

#### 4.2.2. Mitochondria and Parkinson’s Disease

Mitochondria have been found to be implicated in PD in different ways. Similarly to AD, abnormal mitochondrial morphology has also been observed in the fibroblast of PD patients [299]. Abnormal mitochondrial morphology has also been shown using PD cybrid cell lines and mouse models [300,301]. Genes that are involved in familial PD often include PINK1 and parkin, both of which encode for proteins that play a role in mitochondrial maintenance [302,303,304,305]. Beyond this, oxidative stress due to mitochondrial dysfunction also plays a role in PD.

It has been suggested that oxidative stress is implicated in PD [305]. Altered GSH/GSSG ratio was observed in the substantia nigra of PD patients’ [306], and meta-analysis has also shown that increased 8-hydroxy-2′-deoxyguanosine and decreased glutathione were observed in PD patients compared to controls [307].

The anti-diabetic drug metformin was investigated as a PD therapeutic option as well. Metformin treatment in MPTP-induced PD mice models reduced oxidative stress and improved motor movements [308,309]. Similarly, the use of metformin in haloperidol-treated mice was able to reduce oxidative stress and improve catalepsy, but it was not found to significantly improve coordination and balance [310]. Metformin was also able to reduce dyskinesia in mice with rotenone-induced parkinsonism [311]. Furthermore, Metformin could also reduce α-syn and prevent dopaminergic neuronal degeneration in a PD mice model [309]. Moreover, metformin could restore the parkin levels that were observed to be depleted in the substantia nigra of diabetic and high-fat diet mice [312]. Metformin has also been found to be beneficial for PD in cohort studies. Studies have found that type 2 DM patients using metformin had a lower risk of PD as compared to non-metformin users [313,314]. Furthermore, the risk of PD was also decreased in patients using a combination therapy of sulfonylureas and metformin [315]. Currently, there is also a clinical study (NCT05781711) that aims to investigate metformin as a treatment for PD. The effect of metformin on PD could be mediated through the mitochondria. Metformin has been found to normalize mitochondrial function [316] and also improve PD phenotypes under the conditions of mitochondrial dysfunction indicated by hyperactive mitochondria [317]. In addition to that, metformin has also been found to activate AMPK, resulting in several effects such as activation of tyrosine hydroxylase, which is critical for dopamine synthesis, and improving dopaminergic levels and reducing motor impairment [309,318]. However, there have been some studies that have suggested that the effect of metformin does not require AMPK activation [319].

Gliclazide is also potentially beneficial for PD, as a recent study found that gliclazide was able to improve motor and cognitive function in diabetic mice with induced PD [320]. However, meta-analysis has also found that there is a lack of association between sulfonylureas use and PD, suggesting otherwise [321].

The SGLT2 inhibitors canagliflozin and empagliflozin could exert their effect in PD. In a rotenone-induced mice model, a combination treatment of canagliflozin and levodopa/carbidopa was able to improve motor functions accompanied with an increase in striatal dopamine levels [322]. The use of empagliflozin in a PD mice model was also able to improve coordination and movement [116]. In two cohort studies with type 2 DM patients, it was found that the use of SGLT2 inhibitors was associated with a lower risk of PD compared to DPP4 inhibitors [87,117]. Both canagliflozin and empagliflozin could be helping PD through improving mitochondrial dysfunction. Studies in mice have found that both canagliflozin and empagliflozin were able to decrease oxidative stress in PD mice models [322,323,324]. Furthermore, the use of canagliflozin was able to improve other characteristics of mitochondrial dysfunction such as improving mitochondrial transmembrane potential [322]. Both canagliflozin and empagliflozin could also be activating the AMPK/SIRT1 pathway, as an increase in AMPK or phospho-AMPK and SIRT1 was also observed [322,325].

### 4.3. Depression

#### 4.3.1. Introduction to Depression

Depression is a mood disorder that results in prolonged periods of depressed mood accompanied with varying symptoms such as loss of concentration, sleep disruption, and low energy levels, which can interfere with daily life [326]. There are several hypotheses and theories that have been used to explain the pathophysiology of depression [327]. One of the more common theories is the monoamine theory, where the depression is thought to be caused by a reduction in the neurotransmitters dopamine, serotonin, and noradrenaline [328].

#### 4.3.2. Mitochondria and Depression

Mitochondria could also play a role in depression. Multiple aspects of mitochondrial dysfunction have been reported to be associated with depression [329]. For example, oxidative stress is commonly observed in depression. A meta-analysis has revealed that patients with depression have lower levels of antioxidants and higher levels of products generated from oxidative damage [330]. Furthermore, oxidative stress has also been observed in rodent models of depression [331,332,333,334].

Antidiabetic drugs potentially have an effect on depression as well. Studies in mice have shown a potential beneficial effect of metformin in depression. Metformin was able to have anti-depressant effects on mice, where the mice showed reduced immobility time in a forced swimming test (FST) and a tail suspension test (TST) [335,336,337]. One possible mechanism of metformin’s effects on depression is through the AMPK pathway, as studies have found that metformin is able to increase phospho-AMPK levels in mice that exhibited lesser depressive behaviors [336,337]. Furthermore, the use of an AMPK inhibitor was able to prevent the anti-depressant effects exhibited by metformin [336]. Cohort studies involving type 2 DM patients have also found that the use of metformin is associated with a lower risk of depression compared to patients that were not taking any anti-diabetic medication [338,339,340]. Other studies have also shown that the incidence of depression was also lower for diabetic patients taking metformin compared non-metformin users [341]. However, one of these studies showed that while a low metformin dose is associated with lower depression risk, high metformin dose was instead associated with higher depression risk [339]. The association of metformin with disease-associated depression was also investigated in several studies. The association of metformin with behavioral and psychological symptoms of dementia (BPSD) was investigated in patients with both type 2 DM and AD, and it was found that metformin was also associated with a lower odds of depression symptoms [342]. Besides that, the effect of metformin on depression was also investigated in patients with polycystic ovary syndrome (PCOS), and the study showed that the women on metformin, together with lifestyle modifications, had lower odds of having major depression than when just performing lifestyle modifications alone [343]. A small-scale clinical trial has also been done to investigate if metformin benefits depression. In this clinical trial, patients with depression and type 2 diabetes mellitus were either treated with metformin or a placebo for 24 weeks, and it was found that metformin was able to significantly improve depressive performance [344]. There are also new clinical trials that aim to study the efficacy of metformin as an adjunct therapy in treating depression for obese patients (NCT06707012) or in youths with major mood disorders [345].

Gliclazide could also be a potential drug for depression as well. A pharmacovigilance study has found that the use of sulfonylureas showed potential antidepressant effects, showing a disproportionality score of less than 1. Furthermore, the use of gliclazide, in particular, also showed a significant decrease in disproportionality score, suggesting a possible anti-depressant effect [346].

Additionally, SGLT2 inhibitors could also possibly benefit depression. Treatment of canagliflozin in chronic unpredictable mild stress (CUMS)-induced rats was able to show anti-depressant effects, including a reduction in anhedonic behaviors and a reduction in FST immobility times [347]. Similar observations were also made for empagliflozin, where it was found to be able to reduce anhedonia and FST mobility times in rats exposed to chronic unpredictable stress or with reserpine-induced depression [348,349,350]. Furthermore, empagliflozin was also able to show anti-depressant effects in the olfactory bulbectomy rat depression model [351]. Beyond rodent models, a propensity score-matched cohort study of type 2 DM patients found that patients who are using SGLT2 inhibitors have a lower risk of depression compared to those using DPP4 inhibitors [352]. Clinical trials also highlighted the potential of empagliflozin as an anti-depressant drug. In a double-blind placebo controlled randomized clinical trial, patients with moderate to severe depression who received empagliflozin + citalopram had lower Hamilton Depression Rating Scale scores than those who received placebo + citalopram [353]. However, clinical trials for canagliflozin did not show the same results, as a clinical trial showed that treatment with canagliflozin did not reduce the occurrence of adverse mental health events [354].

These two SLGT2 inhibitors could improve depression through the mitochondria in several ways. The effects of both canagliflozin and empagliflozin could be modulated through the AMPK pathway; phospho-AMPK levels were increased by canagliflozin in CUMS-induced rats and by empagliflozin in rats with reserpine-induced depression [347,350]. Furthermore, empagliflozin was also able to reduce oxidative stress, as an increase in reduced glutathione levels was detected in rat brains treated with empagliflozin [350,351]. Figure 3 summarizes the effects of the anti-diabetic drugs on mental health.

## 5. Involvement of Mitochondria in the Pathogenesis and Treatment of Obesity

### 5.1. Introduction to Obesity

Obesity is a chronic complex disease that is a widespread and growing international health issue [355] characterized by the excessive accumulation of fat in the body [356]. An overload of energy substrates in obesity often leads to mitochondrial dysfunction, disrupting lipid and glucose metabolism and compromising adipocyte function and systemic energy balance [357].

### 5.2. Mitochondria and Obesity

Mitochondria are vital for several cellular functions, such as ATP production [358], cofactor biosynthesis [359], and the formation of iron-sulfur (Fe-S) proteins [360]. Fe-S clusters serve as essential cofactors for many cellular proteins involved in DNA repair, metabolism [361], and electron transport [362]. The creation of Fe-S proteins is a complex process centered in the mitochondria, involving multiple proteins and steps [363]. When Fe-S clusters are impaired, ATP production declines, leading to lower energy expenditure and increased fat storage, both of which elevate obesity risk [364].

Diets high in fat can induce mitochondrial fragmentation within white adipose tissue, resulting in smaller, less efficient mitochondria that are less capable of optimal energy utilization. This mitochondrial impairment contributes significantly to obesity by disrupting cellular energy production, which alters metabolic pathways and encourages fat accumulation [365]. Several diabetes medications have shown potential for weight management, possibly positively affecting mitochondrial function. Key drugs under investigation for weight loss include (1) metformin, (2) gliclazide, (3) GLP-1 receptor agonists, and (4) SGLT2 inhibitors. Each section discusses the mechanisms of each drug class, emphasizing their roles in weight reduction and examining how they impact mitochondrial efficiency and Fe-S cluster functionality. By targeting pathways that enhance mitochondrial energy efficiency and utilization, these drugs not only improve glucose homeostasis but may also help mitigate the metabolic disruptions associated with obesity. Their influence on mitochondrial energy pathways offers a promising approach to managing obesity through enhanced metabolic and energy balance.

### 5.3. Metformin and Obesity

Metformin is frequently prescribed as a first-line medication in the treatment of type 2 diabetes mellitus. It is well known for its insulin-sensitizing properties, which enhance glucose uptake by cells [366]. Metformin is recognized not only for its efficacy in managing blood glucose levels but also for its potential to induce weight loss. A study found that sustained metformin use emerged as an independent predictor of long-term weight loss within the metformin group [367]. After the masked treatment phase, participants in this group maintained an average weight loss of 6.2% relative to baseline over years 6 to 15, compared to 3.7% in the intensive lifestyle group and 2.8% in the placebo group. In another recent study evaluating weight loss outcomes, metformin therapy was linked to significantly higher odds of maintaining at least a 10% weight loss and a greater degree of treatment persistence after five years [368]. These findings align with prior research, highlighting metformin’s enduring impact on body weight management.

Metformin has been proposed to promote weight loss through multiple mechanisms by influencing energy balance and modulating key metabolic pathways. These mechanisms include increased fat oxidation, appetite suppression, improved insulin sensitivity, and changes in gut microbiota composition [369]. The mitochondrial effects of metformin are multi-faceted, notably involving the activation of adenosine monophosphate-activated protein kinase (AMPK). AMPK activation is a major energy-sensing pathway that enhances fatty acid oxidation and glucose uptake while inhibiting lipid and protein synthesis [273]. AMPK activation by metformin is primarily achieved by reducing electron flow through mitochondrial complex I, thereby reducing ATP production. This reduction leads to an increase in the AMP/ATP ratio, which in turn activates AMPK, thus initiating metabolic pathways that regulate energy balance [198,370]. Additionally, metformin’s activation of AMPK reduces the expression of proteins crucial for lipid storage and droplet dynamics, such as Cidec, Perilipin1, and Rab8a, promoting lipid mobilization and reducing fat accumulation [371]. Metformin’s direct impact on iron-sulfur (Fe-S) clusters is not well documented, although it may impact the formation and functionality of Fe-S clusters, as these clusters are essential for mitochondrial electron transport and various metabolic processes.

The inhibition of mitochondrial complex I by metformin appears to be dose dependent. In a diabetic rat model, a two-week treatment of the diabetic rats with 100 to 300 mg/kg/day of metformin (which are concentrations that far exceed those prescribed for humans) resulted in a dose-dependent reduction in muscle oxidative capacity, with significant impairments ranging from 21% to 48% at higher doses, as demonstrated in both in vivo and ex vivo assessments [372]. At low, pharmacologically relevant concentrations, metformin was shown to block the lysosomal proton pump v-ATPase, triggering CaMKK2 to phosphorylate and activate AMPK at T172 [5]. Evidence suggests that aside from the gut, most organs accumulate metformin at micromolar concentrations when administered therapeutically, revealing mechanisms that do not rely on complex I inhibition [373]. Therapeutic concentrations of metformin may enhance mitochondrial aerobic metabolism, improving cellular energy status. This hormetic effect illustrates how tissue-specific responses to metformin arise from varying tissue concentrations of the drug [87].

Another mechanism that amplifies AMPK activation is partially mediated by metformin-induced upregulation of growth differentiation factor 15 (GDF15). GDF15 is a stress-response cytokine involved in energy balance, inflammation, and cellular stress response pathways. GDF15 binds to the GDNF family receptor α-like (GFRAL) in the hindbrain, and this interaction suppresses food intake and body weight [374]. Apart from appetite suppression, the GDF15-GRAL pathway was also found to enhance energy expenditure in the skeletal muscle of mice during calorie restriction, supporting its contribution towards energy homeostasis [375]. A human trial found that two weeks of metformin treatment was associated with a 2.5-fold increase in mean circulating GDF15 [376]. In mouse models, metformin inhibited mitochondrial function, leading to downstream effects, including increased GDF15 secretion by intestinal cells [377]. Metformin also increases blood GDF15 levels via upregulation in the kidney, which then acts on the area postrema in the hindbrain, further regulating food intake and body weight [378]. Additionally, metformin-stimulated GDF15 upregulation contributes to full AMPK activation in peripheral tissues without requiring the involvement of the central nervous system [379].

In conclusion, metformin demonstrates significant potential as a weight-loss agent through its effects on mitochondrial function and energy balance. By activating AMPK, metformin promotes fatty acid oxidation, reduces lipid synthesis, and suppresses appetite, contributing to sustained reductions in body weight. Its dose-dependent inhibition of mitochondrial complex I decreases ATP production and increases the AMP/ATP ratio, driving AMPK activation. Research highlights that therapeutic concentrations enhance cellular energy status while avoiding the adverse impacts observed at suprapharmacological doses. Additionally, metformin-induced upregulation of GDF15 further amplifies AMPK activation and results in appetite suppression and enhanced energy expenditure, reinforcing its role in weight regulation. These mitochondrial effects highlight metformin’s unique ability to support weight loss alongside its established role in glycemic control, making it a valuable tool for long-term metabolic health.

### 5.4. Gliclazide and Obesity

Gliclazide is a second-generation sulfonylurea (SU) [380] that effectively lowers blood sugar levels by promoting insulin secretion from pancreatic beta cells [4,381,382], improving β cell function, and aiding in achieving glycemic targets in patients who have not responded to metformin alone [383]. Studies have highlighted additional metabolic effects that may play a role in weight regulation [384,385,386,387,388].

Gliclazide has proven effective in promoting weight loss or preventing weight gain in patients with non-insulin-dependent diabetes. In a 36-month study, patients treated with gliclazide saw a notable decrease in weight, dropping from 68.2 kg to 64.0 kg [389]. Likewise, a 3-month trial observed a notable average weight loss of 1.5 kg in obese diabetic patients [390]. Another multi-center study noted weight loss, especially among obese and elderly participants [391]. However, a 30-month long-term follow-up study observed no significant overall change in body weight, with some patients experiencing weight gain and others losing weight [392].

Gliclazide has demonstrated an ability to shield both healthy and cancerous cells from cell death caused by oxidative stress [393]. This protective effect is probably attributed to its antioxidant properties, which reduce the production of reactive oxygen species and restore mitochondrial membrane potential [393]. Gliclazide’s antioxidant properties have also been shown to improve the pathological changes caused by type 2 diabetes mellitus (T2DM) [280]. Gliclazide, often used in combination with metformin for T2DM treatment in particular dosages [394], was shown to be useful in the treatment of obesity-induced infertility [395].

Iron deficiency is linked to obesity and liver fat accumulation (hepatic steatosis) [396]. However, supplementing with iron can help reduce weight gain and liver fat by enhancing mitochondrial function and increasing the expression of genes involved in heme and iron-sulfur cluster synthesis [396]. Disruptions in the formation of iron-sulfur clusters or in maintaining iron balance can result in various disorders, including complications related to obesity and other serious human diseases [397,398,399]. It has been shown that the depletion of the iron-sulfur cluster (ISC) machinery supporting mitochondrial Fe-S protein assembly leads to inadequate Fe-S and mitochondrial dysfunction [400,401]. In vitro studies suggest that the accumulation of unspecified reactive oxygen species (ROS) dependent on frataxin (FXN), a nucleus-encoded mitochondrial protein, important in Fe-S cluster biogenesis [402], is influenced by a simultaneous reduction in antioxidants in frataxin-deficient conditions [403]. Gliclazide’s antioxidant properties could be of use in treatment plans for obesity and revival of mitochondrial function.

### 5.5. Glucagon-like Peptide-1 Receptor Agonists (GLP-1) and Obesity

Targeting the incretin system is another therapeutic approach used in managing type 2 diabetes. Incretins are hormones released by the intestinal lining in response to eating, which boost insulin secretion and help lower blood sugar levels [404]. GLP-1 receptor agonists are agents that mimic the action of endogenous GLP-1 by binding to and activating GLP-1 receptors located throughout the body, including pancreatic beta cells, the central nervous system, and the gastrointestinal tract [405]. Apart from their anti-diabetic properties, GLP-1 agonists also exhibit anorexigenic properties, which induce weight loss. Of two glucose-lowering incretin hormones, GLP-1 but not glucose-dependent insulinotropic polypeptide (GIP) was found to reduce gastric emptying and exert additional weight loss benefits [406]. Two GLP-1 agonists, specifically liraglutide (in 2014) and semaglutide (in 2021), have been FDA-approved as pharmacologic treatments for overweight status and obesity [407]. When added to lifestyle interventions, the average weight loss difference with GLP-1 receptor agonists, compared to a placebo, ranged from 4% to 6.2% in patients with diabetes, as compared to 6.1% to 17.4% in individuals without diabetes [408]. Patients taking semaglutide were also found to have greater odds (Odds ratio 1.81) of maintaining at least a 10% weight loss after 5 years [368]. However, the odds were lower than those on metformin, topiramate, or bupropion [368].

Weight loss induced by GLP-1 agonists results in reduced caloric intake and alterations in energy metabolism [409]. GLP-1 has been shown to alter central nervous system pathways to suppress appetite and delay gastric emptying, leading to increased satiety [405,409]. These effects are mediated through GLP-1 receptor activation in the hypothalamus, where they influence neural circuits that regulate hunger and satiety, as well as peripheral tissues, where they modulate lipid and glucose metabolism [410]. This dual role of GLP-1 RAs in improving metabolic health and reducing body weight has spurred further research into their underlying mechanisms of action, particularly their impact on cellular bioenergetics.

Emerging evidence suggests that GLP-1 agonists may influence mitochondrial function, a critical regulator of energy homeostasis, and that these effects may contribute to their weight-reducing properties. GLP-1 RAs enhance mitochondrial function by stimulating mitochondrial biogenesis via the upregulation of sirtuin pathways, specifically SIRT1 and SIRT3 in human adipocytes and SIRT1 in mice [411,412]. This enhancement increases energy expenditure, contributing to improved metabolic efficiency. In mice, GLP-1 regulated the expression of microRNAs and enhanced the AMPK activation [412]. This collectively decreases the production of reactive oxygen species (ROS) and enhances antioxidant defenses to reduce lipotoxicity and improve mitochondrial health [412]. The activation of the AMPK pathway also serves to inhibit hepatic lipogenesis [413] Finally, GLP-1 agonist anti-inflammatory benefits contribute to the preservation of mitochondrial function by reducing the infiltration of pro-inflammatory macrophages in adipose tissues and decreasing the expression of inflammatory cytokines such as IL-6 and TNF-α [414]. These mechanisms collectively enhance mitochondrial function, increase energy expenditure, and reduce fat mass, thereby contributing to the weight loss effects observed with GLP-1 RAs.

In summary, targeting the incretin system, particularly through the use of GLP-1 receptor agonists, represents a multifaceted therapeutic approach for managing type 2 diabetes and obesity. Beyond their glucose-lowering effects, GLP-1 agonists exhibit potent weight-reducing properties by suppressing appetite, delaying gastric emptying, and altering energy metabolism. Their impact extends to enhancing mitochondrial function, reducing oxidative stress, and mitigating inflammation, thereby improving metabolic efficiency. These findings highlight the dual metabolic and weight-loss benefits of GLP-1 agonists, underscoring their potential as a cornerstone in the treatment of metabolic disorders. Further exploration of their underlying mechanisms promises to expand our understanding of energy homeostasis and advance therapeutic strategies for obesity management.

### 5.6. Sodium-Glucose Cotransporter Protein-2 (SGLT-2) Inhibitors and Obesity

Sodium-glucose cotransporter-2 (SGLT-2) inhibitors were first identified as anti-diabetic drugs [415] and are now recommended both by diabetes [416] and heart failure [417] guidelines. SGLT2 inhibitors inhibit the function of the SGLT2 receptor, thereby obstructing the reabsorption of glucose in the kidney’s proximal tubule [418]. This leads to glucosuria and a reduction in serum hyperglycemia associated with diabetes [419]. However, inhibiting the SGLT2 also induces natriuresis, or osmotic diuresis, which lowers blood pressure and increases hematocrit [420]. Clinical trials have demonstrated that SGLT2 inhibitors are effective in decreasing the likelihood of cardiovascular incidents, minimizing hospital admissions due to heart failure, and retarding the advancement of diabetic kidney disease in individuals diagnosed with type 2 diabetes [421].

SGLT2 inhibitors boost mitochondrial efficiency by facilitating ketone body oxidation and enhancing ATP hydrolysis [134]. They also control mitochondrial dynamics, decrease oxidative stress, and mitigate inflammation [422]. These drugs alleviate mitochondrial dysfunction, enhance bioenergetics, and uphold ion homeostasis in cardiovascular tissues [423]. Additionally, SGLT2 inhibitors regulate sodium metabolism, lowering intracellular sodium overload and boosting mitochondrial energetics [134]. The mitochondrial benefits of SGLT2 inhibitors play a role in their cardioprotective effects, potentially lowering the risk of diabetic complications [424].

SGLT-2 inhibitors have demonstrated potential in managing weight for individuals with obesity, even in the absence of diabetes. Numerous studies have shown significant weight loss effects compared to placebo, with reductions ranging from 1 to 3 kg [425,426]. A systematic review concluded that SGLT-2 inhibitors are effective for weight loss in non-diabetic obese patients [427]. Meta-analyses have validated these findings, showing significant reductions in body weight and BMI [426,427]. The process of weight loss occurs because the kidneys reduce glucose reabsorption, leading to a loss of calories in urine [425]. Although SGLT-2 inhibitors improved fasting blood glucose levels, their effects on other cardiometabolic parameters, such as waist circumference, blood pressure, and lipid profiles, were less consistent [426,428]. SGLT-2 inhibitors can cause statistically significant reductions in body weight and BMI; however, they do not significantly impact waist circumference in overweight or obese adults who do not have diabetes [428]. Overall, SGLT-2 inhibitors appear to be a promising adjunct therapy for obesity management in non-diabetic individuals.

### 5.7. Discussion for Obesity

Metformin promotes fatty acid oxidation, reduces lipid synthesis, and suppresses appetite, contributing to sustained reductions in body weight. Gliclazide, a sulfonylurea, lowers blood sugar levels by promoting insulin secretion from pancreatic beta cells. SGLT2 inhibitors mainly promote weight reduction by increasing the excretion of glucose in urine, whereas GLP-1 receptor agonists operate by lowering appetite and diminishing food consumption [429]. Combining these treatments could produce synergistic effects, resulting in more consistent weight loss outcomes [430] (Figure 4).

## 6. Conclusions

Antidiabetic agents like metformin, GLP-1 receptor agonists, SGLT2 inhibitors, and gliclazide influence mitochondrial function through mechanisms such as enhancing biogenesis, promoting mitophagy, reducing oxidative stress, and modulating mitochondrial dynamics (Table 1). These actions extend beyond glucose control, offering benefits like improved cellular health, reduced inflammation, and protection against aging-related diseases.

## Figures and Tables

**Figure 1 ijms-26-00364-f001:**
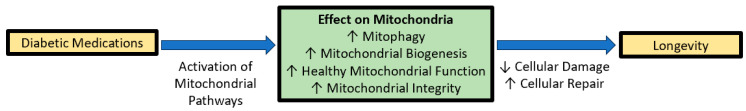
Diabetic medications and the mitochondria.

**Figure 2 ijms-26-00364-f002:**
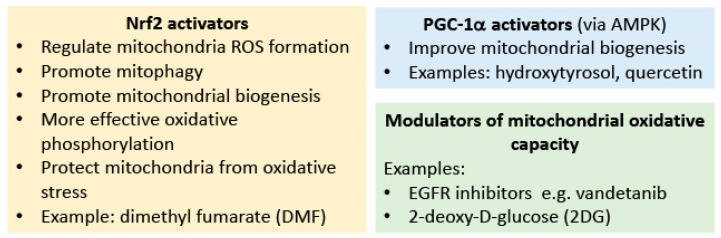
Mode of action of existing drugs that have potential to reverse viral-induced metabolic changes in the mitochondria and hence are candidates targeting treatment of mitochondrial dysfunction in long COVID.

**Figure 3 ijms-26-00364-f003:**
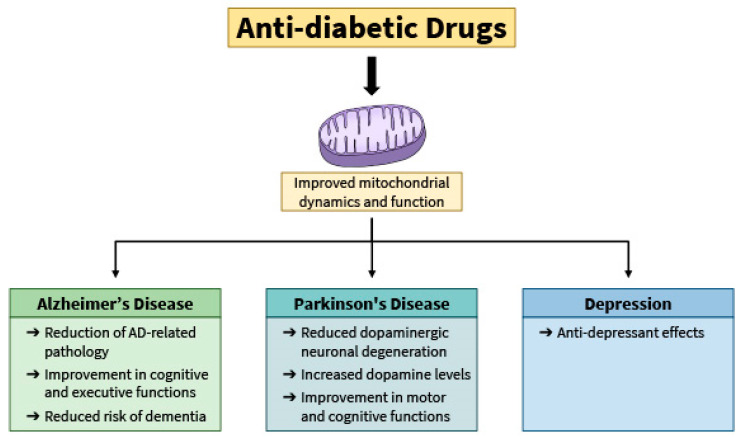
How anti-diabetic drugs improve mental health by modulating mitochondrial dynamics and function.

**Figure 4 ijms-26-00364-f004:**
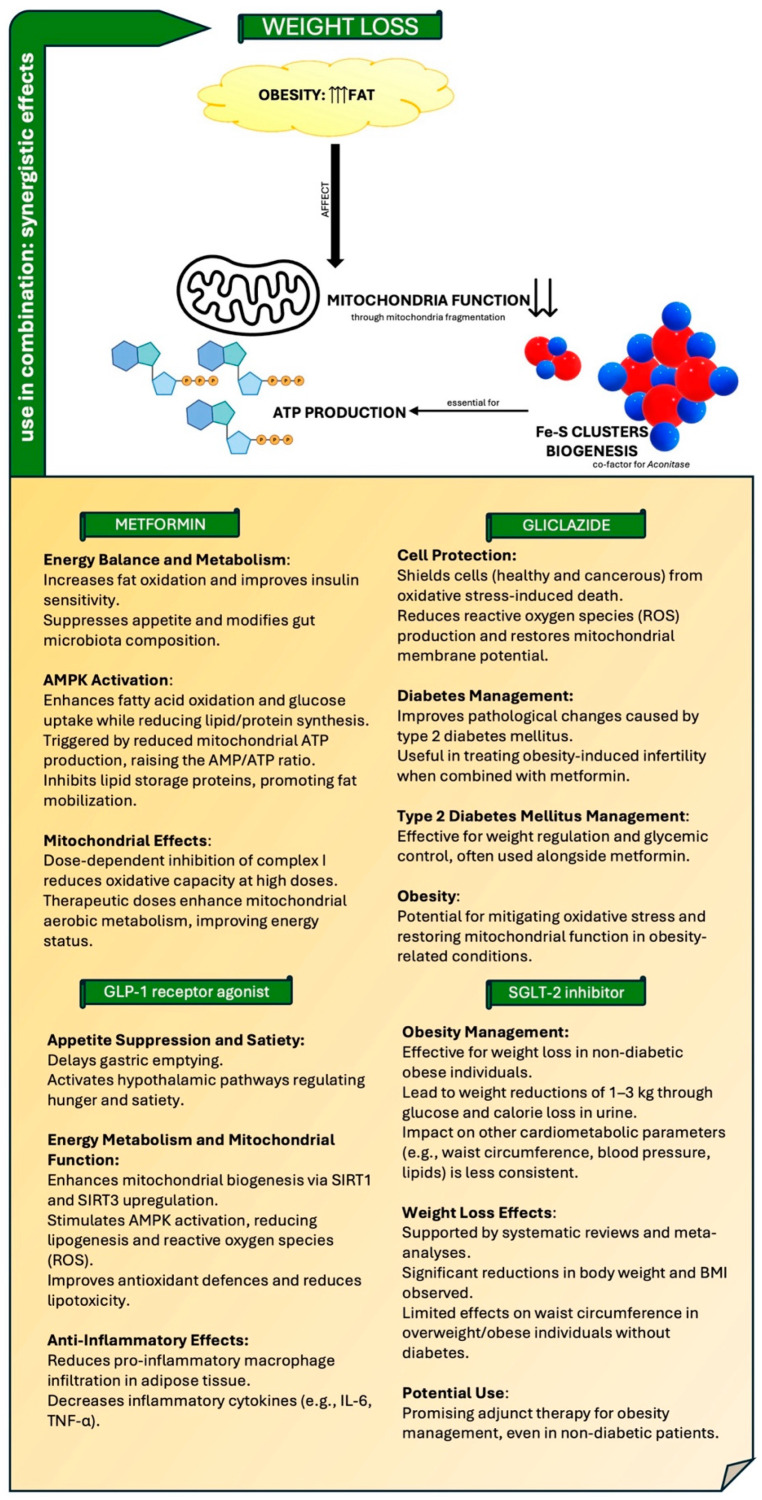
Diagram summarizing how metformin, gliclazide, GLP-1 receptor agonists, and SGLT-2 inhibitors could be used in combination to produce synergistic effects, resulting in more consistent weight loss outcomes.

**Table 1 ijms-26-00364-t001:** Summary of the various antidiabetic drugs and their roles in modulating mitochondria.

Antidiabetic Drug	Mechanism of Action	Role in Modulating Mitochondria	Additional Benefits
Metformin	Inhibits mitochondrial complex I, activating AMPK by increasing AMP/ATP ratio.	Enhances mitochondrial biogenesis and mitophagy.	Reduces oxidative stress by lowering ROS production and increasing antioxidant capacity. Promotes mitochondrial fission and repair. Mimics caloric restriction benefits. Improves insulin sensitivity. Reduces inflammation and aging markers. Reduces risk of long COVID and age-related diseases.
GLP-1 receptor agonists (e.g., liraglutide, semaglutide)	Activates GLP-1 receptors to enhance glucose metabolism.	Increases mitochondrial biogenesis and improves mitochondrial function. Reduces oxidative stress and protects mitochondrial DNA.	Promotes weight loss by reducing appetite and delaying gastric emptying. Lowers cardiovascular and renal risk.
SGLT2 inhibitors (e.g., empagliflozin, canagliflozin)	Inhibits glucose reabsorption in renal tubules; induces caloric loss.	Promotes mitochondrial biogenesis and mitophagy. Enhances mitochondrial dynamics by balancing fission and fusion. Reduces oxidative stress and mitochondrial dysfunction.	Mimics caloric restriction benefits. Reduces inflammation, oxidative stress, and cellular aging. Lowers cardiovascular and renal disease risks.
Gliclazide	Stimulates insulin secretion from pancreatic beta cells.	Improves mitochondrial membrane potential and reduces oxidative stress. Protects mitochondrial function through antioxidant effects.	May aid in weight regulation. Exhibits neuroprotective effects in animal models.

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
