# Peer review of "Mitochondria and the Repurposing of Diabetes Drugs for Off-Label Health Benefits"

_ijms, 2025, doi:10.3390/ijms26010364_

Round 1
Reviewer 1 Report
Comments and Suggestions for Authors
The manuscript titled ‘Mitochondria and the Repurposing of Diabetes Drugs for off-label Health Benefits’ discusses the important information regarding the role of mitochondria in the repurposing of the anti-diabetes drugs for additional clinical benefits beyond diabetes, including unhealthy aging, long COVID, Alzheimer’s disease, Parkinson’s disease, depression, and obesity. This review manuscript is well-written and easy to follow. I only have minor concerns for this manuscript as below:
1. Line 36: “When we eat the sugar glucose, the levels of glucose…”. Since, glucose is a type of sugar, the word ‘sugar glucose’ seems redundant. The authors could consider changing it to “when we eat sugar, the levels of glucose...”.
2. Line 38: “a peptide hormone produced by the b-cells of the pancreas,…”. For clarity, authors are suggested to change b-cells to β-cells (beta cells).
3. Lines 48-49: “T1DM has to be treated by the direct injection of insulin into the blood stream, while…”. This statement is partially true. Although, T1DM is generally treated with insulin, it is not administered by direct injection into the bloodstream. Insulin is typically administered subcutaneously which allows the insulin to be absorbed gradually/in controlled manner into the bloodstream, to mimic the natural release of insulin by the pancreas. Please revise this in the manuscript.
4. There are two sections labeled 2.2., please correct this.
5. Line 172: “ROS are key determinants of aging”. Since this is the first time ROS abbreviation is used in the manuscript, please mention the full form (reactive oxygen species) in parentheses to introduce the term.
6. Line 337: Please add the full form of mtROS.
7. Line 376-377: You described two major clinical trials, Metformin in Longevity Study (MILES) and Targeting Aging with Metformin (TAME) investigating the anti-aging effects of metformin in non-diabetic populations. Are there other clinical trials using SGLT2 inhibitors and GLP-1 receptor agonists? Please mention briefly in this paragraph.
8. Line 493: Ab is not a scientific terminology for beta-amyloid and is abbreviated generally as Aβ. Please correct this at relevant places throughout the manuscript, and in line 656.
9. Lines 692-694: “In terms of…use of metformin and SGLT2 inhibitors.. [295]…”. The study referenced here used incretins and SGLT2-I but not metformin. Please correct this information.
10. Line 740: “..gliclazide was able to motor and cognitive function in diabetic mice induced with PD”. Do you mean gliclazide was able to ‘improve’ motor and cognitive function in diabetic mice induced with PD”. Please revise.
11. Some typo issues: Line 808: “c canagliflozin” should be “canagliflozin”, Line 774: “beneficial effect on metformin” should be “beneficial effect of metformin”, Line 718: “PD mice model in reduced oxidative stress” should be “PD mice model reduced oxidative stress”. Please revise this.
Author Response
Comment 1: Line 36: “When we eat the sugar glucose, the levels of glucose…”. Since, glucose is a type of sugar, the word ‘sugar glucose’ seems redundant. The authors could consider changing it to “when we eat sugar, the levels of glucose...”.
Response 1: Done
Comment 2: Line 38: “a peptide hormone produced by the b-cells of the pancreas,…”. For clarity, authors are suggested to change b-cells to β-cells (beta cells).
Response 2: Done
Comment 3: Lines 48-49: “T1DM has to be treated by the direct injection of insulin into the blood stream, while…”. This statement is partially true. Although, T1DM is generally treated with insulin, it is not administered by direct injection into the bloodstream. Insulin is typically administered subcutaneously which allows the insulin to be absorbed gradually/in controlled manner into the bloodstream, to mimic the natural release of insulin by the pancreas. Please revise this in the manuscript.
Response 3: Done
Comment 4: There are two sections labeled 2.2., please correct this.
Response 4: Done
Comment 5: Line 172: “ROS are key determinants of aging”. Since this is the first time ROS abbreviation is used in the manuscript, please mention the full form (reactive oxygen species) in parentheses to introduce the term.
Response 5: Done
Comment 6: Line 337: Please add the full form of mtROS.
Response 6: Done
Comment 7: Line 376-377: You described two major clinical trials, Metformin in Longevity Study (MILES) and Targeting Aging with Metformin (TAME) investigating the anti-aging effects of metformin in non-diabetic populations. Are there other clinical trials using SGLT2 inhibitors and GLP-1 receptor agonists? Please mention briefly in this paragraph.
Response 7: We are not aware of longevity clinical trials with SGLT2 inhibitors and GLP-1 receptor agonists.
Comment 8: Line 493: Ab is not a scientific terminology for beta-amyloid and is abbreviated generally as Aβ. Please correct this at relevant places throughout the manuscript, and in line 656.
Response 8: Done
Comment 9: Lines 692-694: “In terms of…use of metformin and SGLT2 inhibitors.. [295]…”. The study referenced here used incretins and SGLT2-I but not metformin. Please correct this information.
Response 9: Done
Comment 10: Line 740: “..gliclazide was able to motor and cognitive function in diabetic mice induced with PD”. Do you mean gliclazide was able to ‘improve’ motor and cognitive function in diabetic mice induced with PD”. Please revise.
Response 10: Done
Comment 11: Some typo issues: Line 808: “c canagliflozin” should be “canagliflozin”, Line 774: “beneficial effect on metformin” should be “beneficial effect of metformin”, Line 718: “PD mice model in reduced oxidative stress” should be “PD mice model reduced oxidative stress”. Please revise this.
Response 11: Done
Reviewer 2 Report
Comments and Suggestions for Authors
This manuscript is well written. However, the following issues to be addressed by authors.
1. Auhtors have discussed the role of various antidiabetic agents. Addition of a table with all the possible mechanisms of these antidiabetic agents and their role in modulating mitochondria would be appreciated.
2. There must be a conclusion about this review at the end.
3. Discuss the novelty of the present manuscript.
4. Write the limitations of the present work.
5. Addition of one or two images that show various molecular mechanisms of antidiabetic drugs representing mitochondria would be more advantageous.
6. Section 4.1.1 and 4.1.2: Several chemicals inhibited diabetes and Alzheimer's disease. Authors can refer and cite the following article https://doi.org/10.1002/ptr.8122.
7. At the end of the introduction section, Please discuss the motivation for this review and what novel things this article covers.
8. Metformin is known to cause mitochondrial complex 1 inhibition. Please discuss it.
9. Discuss or make a table on adverse outcomes of antidiabetic drugs, pertaining to mitochondria.
Author Response
Comments and Suggestions for Authors
This manuscript is well written. However, the following issues to be addressed by authors.
Comment 1: Authors have discussed the role of various antidiabetic agents. Addition of a table with all the possible mechanisms of these antidiabetic agents and their role in modulating mitochondria would be appreciated.
Response 1: Done (Table 1)
Comment 2: There must be a conclusion about this review at the end.
Response 2: Done (lines 1072-1078)
Comment 3: Discuss the novelty of the present manuscript.
Response 3: Done (lines 82-84)
Comment 4: Write the limitations of the present work.
Response 4: Done (lines 84-85)
Comment 5: Addition of one or two images that show various molecular mechanisms of antidiabetic drugs representing mitochondria would be more advantageous.
Response 5: Done (Figures 2 and 3 are new)
Comment 6: Section 4.1.1 and 4.1.2: Several chemicals inhibited diabetes and Alzheimer's disease. Authors can refer and cite the following article https://doi.org/10.1002/ptr.8122.
Response 6: This reference is about the health benefits of cruciferous vegetables. These are outside the scope of this review, which is on the repurposing of FDA-approved diabetes drugs.
Comment 7: At the end of the introduction section, Please discuss the motivation for this review and what novel things this article covers.
Response 7: Done (lines 82-84)
Comment 8: Metformin is known to cause mitochondrial complex 1 inhibition. Please discuss it.
Response 8: Done (lines 164-169)
Comment 9: Discuss or make a table on adverse outcomes of antidiabetic drugs, pertaining to mitochondria.
Response 9: There are no adverse outcomes as these are FDA-approved antidiabetic drugs.
Round 2
Reviewer 2 Report
Comments and Suggestions for Authors
Authors have addressed all the issues satisfactorily